# Expression of Cell-Adhesion Molecules in *E. coli*: A High Throughput Screening to Identify Paracellular Modulators

**DOI:** 10.3390/ijms24129784

**Published:** 2023-06-06

**Authors:** Jay Rollins, Tyler Worthington, Allison Dransfield, Jordan Whitney, Jordan Stanford, Emily Hooke, Joseph Hobson, Jacob Wengler, Sandra Hope, Dario Mizrachi

**Affiliations:** 1Department of Cell Biology and Physiology, College of Life Sciences, Brigham Young University, Provo, UT 84602, USA; jay.t.rollins@gmail.com (J.R.); tyworthington@sbcglobal.net (T.W.); allison.dransfield@gmail.com (A.D.); jordanwhit@gmail.com (J.W.); jordanstanford56@gmail.com (J.S.); emily.hooke@gmail.com (E.H.); josephbhobson@gmail.com (J.H.); jcwengler@gmail.com (J.W.); 2Department of Microbiology and Molecular Biology, College of Life Sciences, Brigham Young University, Provo, UT 84602, USA; sandra_hope@byu.edu

**Keywords:** cell-adhesion molecules, paracellular permeability, tight junctions

## Abstract

Cell-adhesion molecules (CAMs) are responsible for cell–cell, cell–extracellular matrix, and cell–pathogen interactions. Claudins (CLDNs), occludin (OCLN), and junctional adhesion molecules (JAMs) are CAMs’ components of the tight junction (TJ), the single protein structure tasked with safeguarding the paracellular space. The TJ is responsible for controlling paracellular permeability according to size and charge. Currently, there are no therapeutic solutions to modulate the TJ. Here, we describe the expression of CLDN proteins in the outer membrane of *E. coli* and report its consequences. When the expression is induced, the unicellular behavior of *E. coli* is replaced with multicellular aggregations that can be quantified using Flow Cytometry (FC). Our method, called iCLASP (inspection of cell-adhesion molecules aggregation through FC protocols), allows high-throughput screening (HTS) of small-molecules for interactions with CAMs. Here, we focused on using iCLASP to identify paracellular modulators for CLDN2. Furthermore, we validated those compounds in the mammalian cell line A549 as a proof-of-concept for the iCLASP method.

## 1. Introduction

The cytoskeleton is a structure that helps cells acquire and maintain their shape and internal organization, providing additional mechanical support that enables cells to carry out essential functions such as division and movement [1]. The cytoskeleton is a multi-component structure working in unison to achieve homeostasis [2]. Prokaryotes and Eukaryotes rely on their cytoskeleton for cell morphogenesis, cell division, DNA partitioning, and cell motility [1,3]. Many bacteria found in nature show morphological polarity [4]. As an example, a rod-shaped bacterium such as *Escherichia coli* (*E. coli*) requires an actin-like polymer formed from MreB to be assembled in order to define its shape [5]. Recently, the discovery that eukaryotic cytoskeletal proteins have homologues in bacteria that affect the cell shape has been a great step forward in understanding cell polarity [3]. In the absence of extracellular contacts, single epithelial cells exhibit very few of the structural characteristics of polarized cells [6,7]. Extracellular contacts between single cells are sufficient to initiate the segregation of membrane and cytoskeletal proteins [8]. Interactions between endothelial or epithelial cells are specified by cell adhesion proteins and result in the formation of paracellular barriers that isolate tissues and organs, creating homeostatic environments [9,10].

Cell adhesion molecules (CAMs) are cell surface proteins involved in establishing cell–cell, cell–extracellular matrix (ECM), or cell–pathogens interactions in a process called cell adhesion [11,12]. In the classical sense, CAMs are grouped into four major families: selectins, the immunoglobulin superfamily (IgSF), integrins, and cadherins [13,14]. Their adhesion properties are relevant in physiological and pathophysiological events [15]. Epithelial and endothelial cell–cell contacts are needed for homeostasis; intercellular junctional complexes are key for their maintenance [10]. Cadherins in the adherens junctions (AJs) provide essential adhesive and mechanical properties [16]. Recently, novel integral membrane proteins, claudins (CLDNs) and occludin (OCLN), have been identified as major membranal components of the tight junction (TJ) [17]. The TJs occupy the most apical region of the cell and create an almost impenetrable barrier that forms without interruption through the entire perimeter of the cell [18,19,20]. AJs and TJs play essential roles in vascular integrity but only the TJ controls the paracellular permeability [21]. CLDNs comprise a multigene family of 4-α-helix transmembrane domains, each member approximately 23 kDa [19,22]. OCLN is a unique protein of the TJ with a structural homology to CLDNs’ 4-α-helix transmembrane domains but it differs in the extracellular loops’ composition and the length and function of its intracellular domains [18,22]. Junctional adhesion molecules (JAMs), another membrane component of the TJ, are members of the IgSF [23]. These non-classical CAMs’ involvement in developmental and physiological processes [24,25], or pathophysiological events such as tumorigenesis and metastasis [26,27], enhances their relevance in translational solutions [28].

The blood–brain barrier (BBB), for example, is a cellular obstacle that maintains the homeostasis of the neural microenvironment [29]. The TJs between brain capillary endothelial cells greatly limit molecular traffic across the paracellular route, with the exception of small molecules (<400 Da) and gaseous molecules [30]. These paracellular and transcellular barrier properties of the BBB, therefore, result in challenges for drug delivery to the central nervous system (CNS), though it has been found that TJs’ dysfunction in the BBB is responsible for increased permeability [31], with CLDN5 being the most relevant CAM in the BBB. CLDN5 knockout experiments in mice result in increased permeability [32]. The double knockdown of CLDN5 and OCLN facilitates enhanced permeation of tracer molecules between 3 and 10 kDa [33].

Other CLDNs are ion permeable [19]. CLDN2 is a paracellular channel-forming protein that is highly expressed in hepatocytes and cholangiocytes, where it regulates paracellular cation and water flow required for the proper regulation of bile composition [34]. Dysregulation of this process increases susceptibility to cholesterol gallstone disease in mice [35]. CLDN10 constitutes paracellular anion channels, and different pathogenic variants of the CLDN10 gene demonstrate dysfunction of the kidney, exocrine glands, and skin [36]. With all the CAMs working together, the TJs create a regulated paracellular barrier that controls the passage of water, solutes, and cells [37,38]. Thus, controlling the TJ could result in an effective strategy for the manipulation of the paracellular permeability [32]. This has been done in a few cases, where modulation of the TJ to prevent viral infections or increase the BBB permeability have been achieved in vitro by peptides corresponding to the extracellular loops of CLDNs [39,40,41]. To the best of our knowledge, Focused Ultrasound and Micro bubbling is the closest technique to FDA approval for drug delivery through the paracellular route [42]. Nevertheless, the current clinical trials are primarily related to Parkinson’s disease [43].

Using a synthetic biology approach, we engineered *E. coli* to recombinantly express CLDN and OCLN in its outer membrane. Overexpression of these proteins causes the unicellular *E. coli* to aggregate due to the adhesion properties of these proteins. Such aggregates can be identified using Flow Cytometry (FC) and the extent of aggregation correlates to the cell–cell adhesion power of individual CAMs. Our method, named iCLASP (inspection of cell-adhesion molecules aggregation through FC protocols) is a high throughput solution to identify paracellular modifiers to potentiate drug delivery of hydrophilic molecules.

## 2. Results and Discussion

### 2.1. Conceptual Model

TJs are intercellular junctions critical for building the tissue barrier and maintaining cellular polarity [44,45]. Recent findings have uncovered CLDN-independent aspects of the TJ’s structure and function, highlighting additional players including JAMs, membrane lipids, mechanical force, and phase separation of the zonula occludens (ZO) family of adaptor proteins (AP), which have all been shown to play important roles in the TJ’s structure and function [46].

TJs have three distinct functions: (1) Barrier: form a permeability barrier that restricts free diffusion of molecules across the intercellular space; (2) Fence: act as a membrane fence that restricts intermixing of apical and basolateral plasma membrane domains; (3) Signaling: form bidirectional signaling platforms that receive signals from the cell interior which regulate the TJs’ assembly and function, and also transduce signals to the cell interior to control cell proliferation, migration, differentiation, and survival [38,45,46].

The first two functions, Barrier and Fence, are achieved through protein–protein interactions (PPIs). The Fence function requires the interaction of cytoskeleton-adapter proteins (ZO)-CLDN (or OCLN or JAM). A recent article suggested that Barrier and Fence functions are independent of each other, indicating that in vivo ZO-1 is dispensable for the Barrier function [47]. The Barrier function can be impacted by the use of CLDN extracellular loop-peptides or other protein engineering strategies employing CLDNs [39,48,49]. We thus conclude that disrupting PPIs of CLDNs equates to disrupting cell–cell adhesion.

As with all noncovalent macromolecular interactions, adhesion molecules bind to each other with equilibrium affinities that are defined by their association and dissociation rates [50,51]. However, the efficiency of cell adhesion is not simply a function of the solution-phase equilibrium affinities of adhesion molecules for one another. Adhesion molecules in the plasma membrane and ECM are limited to two dimensions. Thus, even low-affinity molecular interactions may stabilize adhesion if there is time for sufficient bonds to form along the plane of cell contact. The efficiency of cell adhesion and the resulting strength of adhesion reflect multiple factors that dictate the probability of formation of bonds between adhesion molecules on cell or matrix surfaces [52,53]. Cell adhesion can be further stabilized by events that occur after the initial interactions of CAMs, as salting-in or -out can have a strong impact on cell adhesion as highlighted by the studies of the Hofmeister series [54,55]. Furthermore, cadherins are CAMs that have been found to be calcium-dependent [56,57].

In this report, we designed a strategy that will examine PPIs of the TJ membrane component CLDN and OCLN. As mentioned above, these proteins alone can exhibit cell-adhesion properties in the absence of adapter proteins and mammalian cytoskeleton proteins [47], and manipulating these PPIs corresponds to effects in the permeability of the paracellular barriers [39,49]. Thus, in this current work, we do not seek to explain or analyze all complexities of the TJ (Figure 1). Our system has isolated each membrane protein from a complex environment and equated PPIs as a force that will drive non-native cell–cell interactions in *E. coli*.

### 2.2. Expression System Design

The initial goal of our strategy was to recombinantly express CLDNs or OCLN in the outer membrane (OM) of *E. coli* to examine the consequences of endowing a unicellular organism with cell-adhesion molecules. In the strict sense, only outer membrane proteins (Omp) are expressed in the OM of *E. coli* [59]. For pathogenic strains, many of these Omp also serve as virulence factors for nutrient scavenging and the evasion of host defense mechanisms [60]. Omps are unique membrane proteins in that they have a β-barrel fold and can range in size from 8 to 26 strands, with both N- and C-terminus located in the periplasmic space [59,60].

Our strategy created a fusion protein between OmpW (accession number P0A915) and a human CLDN protein. The N- and C-terminus of OmpW are located in the periplasm; thus, a fusion between OmpW and CLDN, also with N- and C-terminal ends under the plasma membrane will result in an exposure of the adhesive domains (extracellular loops) of CLDNs to the exterior of the cell (Figure 2A).

### 2.3. Flow Cytometry of Bacterial Cells Expressing 4-α-Helix CAMs

BL21 DE3 cells are transformed with the corresponding plasmids (OmpW, OmpW-CLDNs, or OmpW-OCLN). Cells grow to an OD_600_ of 1, induced with 1 mM IPTG, and are allowed to continue growth for 18 h at room temperature. Cells are prepared for a 96-well plates setup. Cell suspension (50 μL cells and 150 μL of PBS) is dispensed per well and runs through a Beckman Coulter Cytoflex flow cytometer (Beckman Coulter, Indianapolis, IN, USA). Readings were collected using the side scatter (SSC) data from the 405 nm (violet) laser for excitation and a 405/10 bandpass filter for emission detection. The violet laser SSC has a greater sensitivity than the forward scatter (FSC) or SSC detection of the 488 nm (blue) laser for detecting alterations in cell shape as reported by others [61] and according to our assessment of flow data results in this study. The SSC area versus height readings were plotted for the data analysis. Cytoflex-generated FCS flow data files were analyzed using FlowJo 10 software (BD Biosciences, Ashland, OR, USA). The violet SSC-area by SSC-height data was gated for the data analysis set to exclude upper and lower extremes that would interfere with the calculation of the slope of the line (Appendix A). Structural cell changes are detected as the area readings move away from the height readings, where area increases at a lower rate than height and the slope of the line decreases (Figure 3). Further manipulation of the data using RStudio 2023.03.1 resulted in Experimental Slopes calculated for each sample (see Materials and Methods) which were used to plot the results.

Prior to concluding that the extracellular loops (ECL) of CLDN1 were responsible for the cell aggregation behavior of BL21 DE3, we prepared a final control. In *E. coli*, disulfide bond formation occurs in the periplasm. DsbB is a membrane protein enzyme that helps catalyze the disulfide bond reaction alongside DsbA and DsbC [62,63,64]. DsbB is a 4-α-helix membrane protein with a topology similar to that of CLDN [65].

Figure 4 represents the properties of a chimeric protein, DsbB-CLDN5, in which the ECLs of CLDN5 replaced those of DsbB (Appendix A). DsbB alone was unable to increase the cell aggregation of *E. coli* by its overexpression on the OM of BL21 DE3. Contrary to that, DsbB-CLDN5 chimera increased cell aggregation as observed by the Experimental Slope calculated for BL21 DE3 cells (Figure 4C).

Endowed with this tool, we proceeded to compare the adhesive properties among TJ proteins of a 4-α-helical topology. CLDNs are a family of 27 proteins in mammals [19]. We used OmpW as a fusion partner to express CLDN1 through 10, OCLN, and tricellulin (TRCL); the latter is the first integral membrane protein found to concentrate at the vertically oriented TJs of tricellular contacts [66]. In the analysis, we included stargazing (STRGZ), an AMPA receptor adapter protein [67], believed to have some structural and functional homology to CLDNs [19]. This homology seems to suggest STRGZ may behave as a CAM. Figure 5 correlates the adhesive properties of the above-mentioned proteins by plotting the Experimental Slope calculated, as previously described. The controls for these experiments are BL21 DE3 cells with the empty plasmid pET28a, and BL21 DE3 cells recombinantly expressing OmpW alone in a pET28a backbone.

Our results highlight the notion that CLDN5, the main CLDN of the BBB, is the strongest cell–cell CAM among the members of this family [31,68]. CLDN5–CLDN5 PPIs have not been measured experimentally. Nevertheless, the Experimental Slope of OmpW CLDN5 was the largest. Additionally, we observed an Experimental Slope above the controls for OmpW-STRGZ, indicating that it has the potential to act as a CAM in the brain [67]. Considering our results reported here for a chimeric DsbB-CLDN5 (Figure 4) and STRGZ, we propose that iCLASP is a favorable method to discover cell-adhesion properties in novel or suspected CAMs.

To determine the extent to which iCLASP can be used to determine changes in TJ-induced paracellular permeability, we resourced to study a dose response of molecules known to non-specifically alter trans-epithelial electrical resistance (TEER) [69]. TEER is a widely accepted quantitative technique to measure the integrity of TJs’ dynamics in in vitro models of endothelial and epithelial monolayers; it is considered as a strong indicator of the paracellular permeability prior to the evaluation for transport of drugs or chemicals [69,70]. Sodium Caprate is a known detergent that disrupts CLDN–CLDN interactions resulting in an increased permeability of the paracellular space [71,72]. On the other hand, Ethanol has been described as an agent that increases CLDN–CLDN interactions, resulting in decreased paracellular permeability [73].

Figure 6 exhibits dose-dependent changes to cell–cell interactions under the effects of Caprate or Ethanol. Thus, considering how iCLASP can identify changes in CLDN–CLDN interactions, mirroring the results of both Caprate and Ethanol in TEER experiments, we suggest it can be used for the discovery of molecules that affect paracellular permeability controlled by CLDN proteins and perhaps other CAMs.

### 2.4. Ion Permeable CLDNs and the Hofmeister Effect

CLDNs are key regulators of barrier properties of the TJ. CLDNs are better recognized for their primary responsibility of tightening the paracellular pathway. A few CLDNs, and, therefore, the TJs they regulate, serve as paracellular channels for water and ions [74]. Among CLDNs with these properties, we have chosen cation-selective human CLDN2 [35] and anion-selective human CLDN10 [36].

The Hofmeister series characterizes ions by their ability as “salt-in” or “salt-out” proteins [75]. Experimental results show that ion cooperativity may play a significant role in affecting water properties [75,76]. In Figure 7, both CLDN2 and CLDN10 are examined under the effects of salts that have been well-characterized in the Hofmeister Series [77,78,79]. We performed the iCLASP experiment with the same salts for CLDN1, CLDN2, and CLDN10 (see Materials and Methods). CLDN1 did not display any trends under the conditions of any of the salts employed; Chloride salts (Al^+3^, Mg^+2^, Na^+^, and Rb^+^) only show an effect with CLDN2 (Figure 7). Sodium salts of different negative anions (HPO_4_^−^, Cl^−^, NO_3_^−^, and ClO_4_^−^) only show an effect with CLDN10 (Figure 7).

For CLDN2, the Hofmeister series for cations has a trend of increasing cell–cell adhesion (Figure 7). Al^+3^ has a more relaxing effect on CLDN2–CLDN2 interactions, while Rb^+^ aids increase CLDN2–CLDN2 PPIs. The Hofmeister series ranks the salts by degree of stabilization or destabilization. In the case of CLDN2, losing stability further drives CLDN2–CLDN2 interactions, which would be detrimental to the cation permeability function, thus tightening the TJ but reducing the ability to sort cations or excluding them from needed filtering. In the kidney, CLDN2 is vital for efficient Na^+^ and water reabsorption in the proximal tubules. Accordingly, the S2 segment of the proximal tubules in claudin-2 KO mice showed significantly reduced net transepithelial reabsorption of Na^+^, Cl^−^, and water [80]. CLDN10, on the other hand, has a trend that seems to indicate that, at lower stability, the permeability of the TJ will increase. Mice with a deletion of CLDN10 in the thick ascending limb impairs paracellular Na+ permeability and leads to hypermagnesemia and nephrocalcinosis [81]. The data provided by KO experiments of CLDN2 and CLDN10 represents the complete absence of these crucial proteins for kidney function. Our experimentation of these protein’s function influenced by salts in the Hofmeister series reveals further information regarding the effects on the TJ’s microenvironment and its effect on the TJ’s barrier function. Taken together, when salts decrease the stability of the microenvironment, CLDN2 loses its function but tightens the TJ (hypopermeability), corresponding to the CLDN2 KO results in which reduced net transepithelial reabsorption of Na^+^, Cl^−^, and water is observed [80]. On the other hand, our data suggests that when CLDN10 loses its function, the TJ’s organization may lead to hyperpermeability, supported by the data suggesting that KO CLDN10 leads to hypermagnesemia and nephrocalcinosis, characterized by the deposition of calcium in the kidney parenchyma and tubules [81].

### 2.5. iCLASP Model

Finally, we propose a graphical model to illustrate the power of iCLASP (Figure 8). Bacterial cells are transformed with plasmids hosting 4-α-helix CAM membrane proteins. CAMs induce multicellular behavior in *E. coli* (aggregates or clumps of cells) that can be measured by Flow Cytometry protocols. A typical experiment will contain cells expressing CAMs, untreated as the internal control accounting for growth and protein expression variability between experiments, and those treated. The power displayed by iCLASP is two-fold; the screen of a library of compounds may identify, in the same experiment, compounds that decrease or increase the size of aggregates. The first will represent cases in which the compound induces hyperpermeability of the paracellular space, while the second represents a hypopermeability outcome. Hyperpermeability may be desirable when trying to overcome, as an example, the BBB for drug delivery. In other cases, TJ dysfunction may contribute to epithelial permeation disorder as is the case of multiple intestinal diseases such as inflammatory bowel diseases (IBD) [82]. In such cases, hypopermeability may be required to foster cell–cell interactions to prevent progression of the disease, inflammation, and recurrent infections [83,84]. In any event, these compounds can also shed light on the function of particular CAMs, their function, and their role in physiology and pathophysiology.

Confident that iCLASP will present an opportunity to identify paracellular modulators, we decided to put it to the test with a 50,000-compound library (ChemBridge, DIVERSet library; see Materials and Methods) against human CLDN2. Figure 8 schematically represents the flow of the work performed.

We expressed plasmid hosting OmpW-CLDN2 in BL21 DE3 cells as described in Materials and Methods. We used 96-well plates with wells 1 and 12 in each row containing no treatment, thus providing 16 wells of control per plate. These control wells identify the Experimental Slope of the cells grown for the experiment, resulting in an accurate state of protein expression and phenotype observed. Each experiment was conducted in quadruplicates. The data for the 50,000-compound library were analyzed to identify the top 40 compounds that increased or decreased the Experimental Slope. These 80 candidates were tested to identify the best compounds, this time on the same day of the experiment (96-well plates in the quadruplicate). From this smaller screen, we identified three compounds in each category. Typical results for single compounds that decreased the Experimental Slope were in the range of 20–35%, while the values of compounds that increased the Experimental Slope of cells expressing OmpW-CLDN2 were between 8–15%.

### 2.6. Paracellular Modulators of CLDN2 in Mammalian Cells A549

CLDN2 has been linked to cell proliferation in vitro in A549 cells. The downregulation of CLDN2 expression decreased proliferation in human lung adenocarcinoma A549 cells [85]. Additionally, CLDN2 binding peptides downregulate its expression and results in anticancer resistance in human lung adenocarcinoma A549 cells [49].

Concluding, using iCLASP, we identified six compounds that later were used in tissue culture experiments to measure cell proliferation in A549 cells in order to validate our findings. Compounds that decreased the Experimental Slope of *E. coli* hosting OmpW-CLDN2, in other words decreasing PPIs, resulted in lower proliferation rates. Compounds that in iCLASP increased Experimental Slopes resulted in the increased proliferation of A549 cells. Combining our results with the previously mentioned literature, the six compounds identified by iCLASP may have an effect in lung cancer biology. A recent report discusses the multifaceted aspects of CLDN2 in cancer. CLDN2 promotes breast cancer liver metastasis by promoting cancer cell survival [86]. The article also discusses that CLDN2 is functionally required for colorectal cancer liver metastasis and that CLDN2 expression in primary colorectal cancers is associated with poor overall and liver metastasis-free survival [86]. The proliferation of human adenocarcinoma A549 cells was decreased by CLDN2 knockdown together with a decrease in the percentage of S phase cells [87]. Gallotannin (Gltn) has been previously demonstrated to have potent anti-tumor properties against cholangiocarcinoma in mice and recently in vitro against triple-negative breast cancer with its effects associated with slowed cell cycle progression and S phase arrest [88]. A recent review [89] examined advances in Gltn anticancer activities and drug delivery systems for efficiency improvement. Recently, a human clinical pilot trial investigated the pharmacokinetics of Gltn-metabolites [90]. As depicted for Gltn, validating hit compounds identified by iCLASP in mammalian cells opens the door for future screening in search of molecules that can modify CAM’s properties and aid in physiological or pathophysiological situations. The compounds tested in Figure 9 (see also Appendix A) will be used to identify First-In-Family and First-In-Class CLDN2 modulators.

The drug discovery of small molecules from the target selection through to the clinical evaluation is a very complex and challenging area of study. The main obstacle in drug discovery is the initial hit-finding [91,92]. The absence of tools to overcome the BBB leads directly to the present situation in neurotherapeutics, where few effective treatments exist for most brain-related disorders [93,94,95]. Intentional drug discovery or drug design is hampered by the lack of knowledge in the fundamental transport properties of the BBB and the molecular and cellular biology of the brain capillary endothelium dictated in great part by the TJ. Currently, dynamics of the BBB and cytotoxicity testing and drug permeation are carried out in vitro using TEER [96]. Some challenges encountered in TEER are the lengthy set-up, up to three weeks prior to obtaining adequate conditions for measurements, and the availability of suitable cell lines that represent the desired system of study [69]. Finally, identified cellular systems may fail to produce adequate and measurable TEER values [69,96]. Prodrug methods used to improve drug penetration via the transcellular pathway have been successfully developed, and some prodrugs have been used to treat patients [97]. The use of transporters to improve drug absorption (e.g., antiviral agents) has also been successful in treating patients. Other methods, including (a) blocking the efflux pumps to improve transcellular delivery and (b) the modulation of cell–cell adhesion in the intercellular junctions to improve paracellular delivery across biological barriers, are still in the investigational stage [98].

## 3. Materials and Methods

### 3.1. Reagents and Genes

For Flow Cytometry, flat bottom 96-well cell culture plates were employed from GeneClone (El Cajon, CA, USA). All genes employed in this study were synthesized by TWIST biosciences (San Francisco, CA, USA) and cloned in pET28a. All salts used for the experimental performance of Figure 5 were obtained from Sigma Aldrich (St. Louis, MO, USA). TEV protease was obtained from New England Biolabs (Ipswich, MA, USA). Accession numbers of all full-length genes used in this study can be found in Appendix A.

### 3.2. Transformation, Cell Growth, and Protein Expression (LB, Fresh Transformations, Controls)

Plasmids (pET28a, pET28a-OmpW-CLDN, pET28a-cpOmpW-JAM-A) are transformed in BL21 DE3 cells; plates are prepared with LB, 2% agar, and 100 μg/mL of kanamycin. A single colony is grown over night in 5 mL of LB and kanamycin. A 1:1000 dilution of the overnight culture is started in the morning. Cells are grown at 30 C until OD_600_ is 1.0. IPTG (1 mM) and is used to induce protein expression; cells are placed in a shaker at 21 °C (room temperature) and allowed to continue growth for 18 h. Samples can be analyzed by Western blot and anti-HIS antibody (ab1187) Abcam (Cambridge, UK).

### 3.3. Flow Cytometry Data Collection and Analysis

Samples were prepared in suspension by mixing 50 µL of cultured cells with 150 µL of PBS and then run through a Beckman Coulter Cytoflex flow cytometer (Beckman Coulter, Indianapolis, IN, USA). Readings were collected using the side scatter (SSC) data from the 405 nm (violet) laser for excitation and a 405/10 bandpass filter for emission detection. The violet laser SSC has a greater sensitivity than the forward scatter (FSC) or SSC detection of the 488 nm (blue) laser for detecting alterations in cell shape as reported by others [61] and according to our assessment of flow data results in this study. SSC area versus height readings were plotted for the data analysis. Cytoflex-generated FCS flow data files were analyzed using FlowJo 10 software (BD Biosciences, Ashland, OR, USA). The violet SSC-area by SSC-height data was gated for the data analysis and set to exclude upper and lower extremes that would interfere with the calculation of the slope of the line. Structural cell changes are detected as the area readings move away from the height readings where area increases at a lower rate than height and the slope of the line decreases. Below is the code used for the RStudio 2023.03.1 analysis (see Box 1 below), generously prepared by Stephen Picollo, Biology Department, Brigham Young University.

Box 1The code used for the RStudio 2023.03.1 analysis.# Load required librarieslibrary(readr)library(ggplot2)library(hexbin) #data path is the folder that the files are in#figures path is the folder of pdf’s that it will generate#slopes path is the excel sheet of slopes it will generate-keep the ‘.csv’data_dir_path = “C:\\Users\\jayroll\\Desktop\\working files”figures_dir_path = “C:\\Users\\jayroll\\Desktop\\7-17”slopes_file_path = “C:\\Users\\jayroll\\Desktop\\7-17.csv” # This will take the saved data and plot it with trendline and save it to a PDF filegetPlot <- function(data, figure_file_path){  myDotPlot <- ggplot(data, mapping = aes(x = VioletSSC.A, y = VioletSSC.H)) +    geom_hex() +    geom_smooth(method = “lm”) +    theme_bw(base_size = 18)  ggsave(figure_file_path, width = 8, height = 6)} # This will get the slope of the trendline and then return the slopegetSlope <- function(data){  model <- lm(VioletSSC.A ~ VioletSSC.H, data=data)  return(coef(model)[[2]])} if (!dir.exists(figures_dir_path))    dir.create(figures_dir_path)csv_file_paths = list.files(path = data_dir_path, pattern = “*.csv”, full.names = TRUE)slopes = NULLfor (csv_file_path in csv_file_paths) {  print(paste0(“Processing”, csv_file_path))  data <- read.csv(csv_file_path, header=TRUE, sep = “,”)  slopes = c(slopes, getSlope(data))  figure_file_path = paste0(figures_dir_path, “\\”, sub(“.csv”, ““, basename(csv_file_path)), “.pdf”)  getPlot(data, figure_file_path)} write_csv(data.frame(File = csv_file_paths, Slope = slopes), slopes_file_path)

Box lines with # indicate the functions of the code. The code uses a working file from FlowJo, generates hexbin plots to visualize analyzed data, and generates a spreadsheet of the calculated slopes as a. csv file. More detailed protocols of the drug discovery process using the ChemBridge DIVERSet library are detailed in Appendix A.

### 3.4. Cell Proliferation Assay

The first day of the proliferation assay, A549 cells (ATCC CCL-185) consisted of 100,000 cells seeded on 48-well plates. On the second day (at approximately 16 h), cells were treated with PBS or compounds to a final concentration of 10 μM. After 24 h, proliferation assays were performed using the ATPlite Luminescence Assay System (PerkinElmer, American Fork, UT, USA), following the manufacturer’s instructions.

### 3.5. Statistical Analysis

Flow Cytometry data (Experimental Slope) were analyzed using SAS software version 9 (SAS Institute Inc., Cary, NC, USA) and the Mixed Procedure method to generate *p*-values, standard deviation, and standard error and to determine statistical significance (for Figure 5). For all experiments, α = 0.05. Data was collected for each sample in four different experiments (*n* = 4). Each condition was measured in 12-replicates. Thus, for each, data points correspond to the average of 12-replicates and *n* = 4, or 48 data points. Statistical differences were identified for all samples in each graph. The final analysis concluded that all treatments are statistically significant (*p* < 0.01) and significantly different from each other and the control (asterisks omitted for display purposes).

### 3.6. ChemBridge DIVERSet Library

The initial HTS of BL21 DE3 cells expressing OmpW CLDN2 was performed using the ChemBridge DIVERSet library (San Diego, CA, USA). The library consisted of 50,000 compounds. The hits identified according to our laboratory nomenclature (Figure 8) are presented by the ChemBridge seven-digit identifier in Appendix A. The chemical structure of the compounds is also shown in Appendix A. More detailed protocols of the drug discovery process using the ChemBridge DIVERSet library are detailed in Appendix A.

## 4. Conclusions

Currently, there are no therapeutic solutions to control the permeability of the paracellular space in the clinical setting. CAMs, like the TJ’s integral membrane proteins, are responsible for paracellular permeability. In this article, we present evidence for a high throughput *E. coli*-based method that recombinantly expresses CAMs. This expression results in bacterial cell aggregates that can be quantified and qualified using Flow Cytometry. The method, named iCLASP, can examine small molecule libraries and identify candidates that increase or decrease permeability. Compared to classical methods, iCLASP is faster, needing only three days to set up (from transformation to experiment), and Flow Cytometry is performed at speeds of one 96-well/h in our laboratory. When iCLASP was used to identify CLDN2 paracellular modulators, it was able to identify six compounds in which effects in bacteria expressing CLDN2 translated to literature reports of effects of CLDN2 on A549 cell proliferation. CAMs are responsible for cell–cell, cell–extracellular matrix, and cell–pathogen interactions. The iCLASP method is a HTS method to examine CAMs and aid in the initial hit-finding down the drug discovery pipeline.

## Figures and Tables

**Figure 1 ijms-24-09784-f001:**
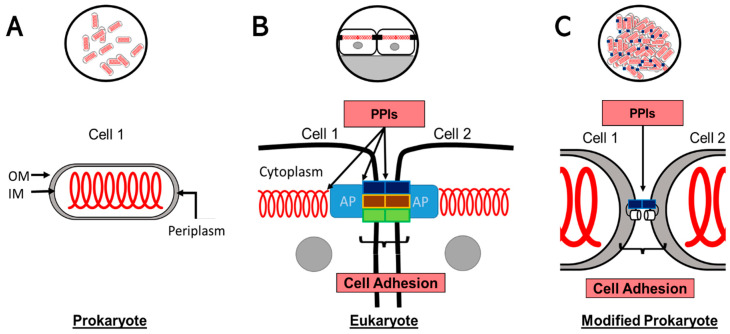
Synthetic biology approach to cell adhesion. (**A**) Depicts a rod-shaped Prokaryote and its cytoskeleton (red). Cytoskeletal filaments such as FtsZ and MreB in the rod-shaped cell are homologs to Eukaryote proteins [3,58]. The insert shows how these unicellular organisms do not establish permanent cell–cell interactions. Outer membrane (OM), Inner membrane (IM), and periplasm are highlighted. (**B**) Two Eukaryote cells interacting through a TJ. PPIs are needed at multiple interfaces of the TJ to reproduce its proper functions (Fence, Barrier, and Signaling). At the membrane level, the TJ is composed of CLDNs (blue box), OCLN (brown box), and JAMs (green box), connected to the cytoskeleton (red spiral) through adapter proteins (AP) such as ZO-1. The insert exemplifies how epithelial cells will use TJs to create cell polarity and barriers. (**C**) Modified Prokaryote. Following our design to fuse CLDN (blue box) and OmpW (white cylinder), *E. coli* cells overexpressing these fused proteins will form non-polarizing pseudo junctions shuttled to the outer membrane (OM). There are no intracellular (periplasmic) PPIs for the fusion protein. The insert demonstrates how, hypothetically, these cells will create large, disordered aggregates of cells without other consequences for cell biology.

**Figure 2 ijms-24-09784-f002:**
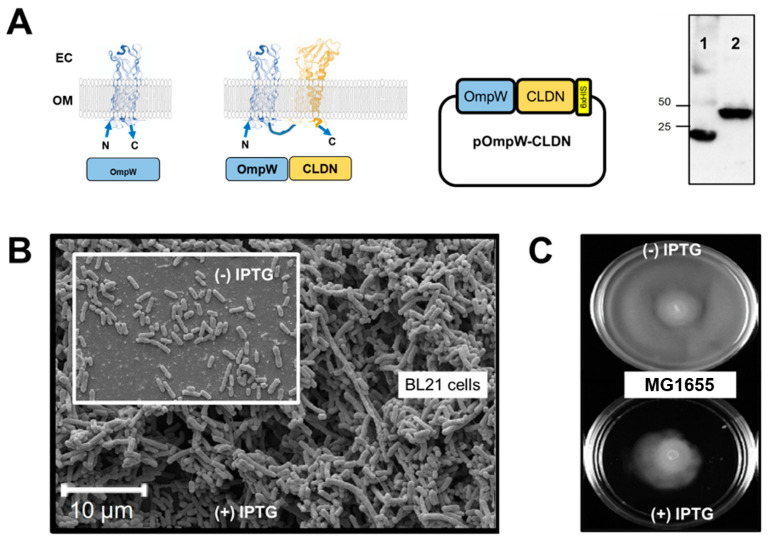
Strategy for claudin (CLDN) expression in *E. coli.* (**A**) OmpW is a native outer membrane protein of *E. coli* that populates the outer membrane; its N- and C-terminus are in the periplasm. These features make OmpW a suitable fusion partner for a 4-α-helix membrane protein such as CLDN, OCLN, and others. Plasmids are synthesized with a C-terminal His-tag for detection. Anti-His antibody Western blot detection of 1-OmpW (24 kDa) and 2-OmpW-CLDN (47 kDa). The Western blot was cropped from a larger image, provided. Appendix A contain amino acid sequences of OmpW and OmpW-CLDN, respectively. (**B**) Consequences of overexpressing CLDN1 in the OM of a Gram-negative Prokaryote. BL21 DE3 cells are captured in Negative Staining experiment Electron Microscopy displaying their natural behavior (unicellular). In the background photo, BL21 DE3 after 18 h of protein expression of OmpW-CLDN1; large aggregates are evident. (**C**) MG1655, a motile variant of *E. coli*, growing on plates of LB + 0.25% agar. Top image contains MG1655 cells after 24 h of growth, from a single drop of cells at OD_600_ = 1, placed at the center of the plate. In contrast, and under identical conditions, when protein expression (OmpW-CLDN1) is induced (1 mM IPTG in the LB+ agar plate), the cells’ horizontal expansion is reduced in a 24-h period. The plate seems to indicate that aggregation of the cells, induced by expression of OmpW-CLDN1, prevents them from displaying their full motility, and thus cannot advance beyond a small radius from the center.

**Figure 3 ijms-24-09784-f003:**
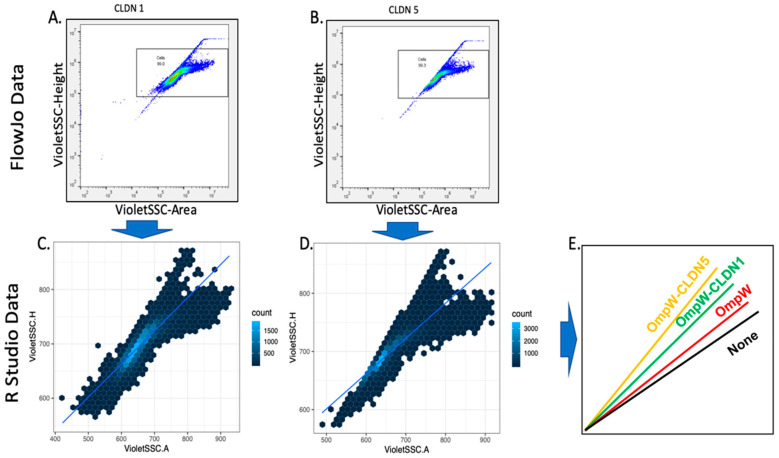
Data analysis using FlowJo and R Studio software. Flow cytometry data were gated in FlowJo software (**panels A**,**B**), and R Studio converted the data into Hexbin graphs (**panels C**,**D**). Experimental Slopes are determined and compared (**panel E**); cells alone and expressing OmpW only display low slopes while CLDN expressing cells have larger slopes. (**Panel E**) graphically demonstrates the results that can be obtained from this analysis.

**Figure 4 ijms-24-09784-f004:**
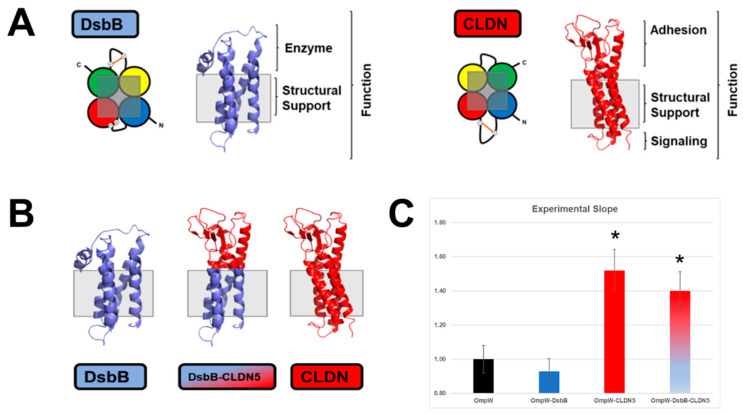
DsbB and ECLs of CLDN5 chimera drive cell-cell interactions. (**A**) Structural make up of DsbB and CLDN5. It highlighted the need for both structural support and active domains to result in a functional unit. (**B**) Graphical depiction of the analyzed proteins, including DsbB-CLDN5 chimera. (**C**) Results of the Flow Cytometry analysis of aggregation of BL21 DE3 cells expressing the corresponding proteins (OmpW, DsbB, OmpW-CLD5, and OmpW-DsbB-CLDN5 chimera). All points are expressed as the average value of 12-replicates in 4 different experiments (*n* = 4) ± SDEV. Asterisks indicate experimental slopes statistically different from OmpW (*p* < 0.01).

**Figure 5 ijms-24-09784-f005:**
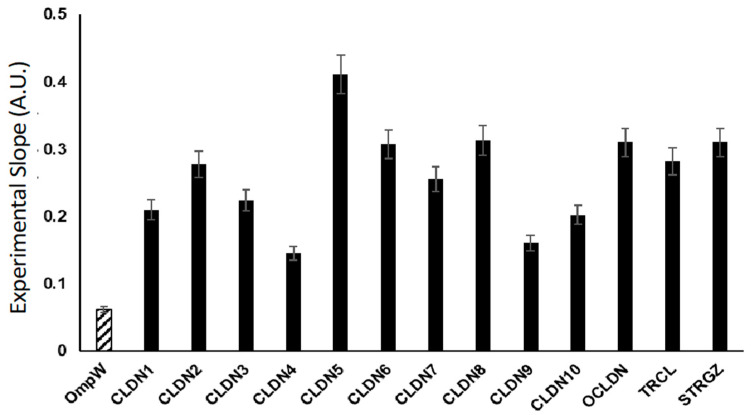
Experimental slopes of 4-α-helical CAMs. Experimental slopes’ differences as a result of protein expression of 4-α-helical CAMs are presented here. The Flow Cytometry analysis enables the determination of Experimental Slopes that represent adhesive properties of CAMs. Data presented here was statistically analyzed after subtracting the value of the Experimental Slope of BL21 DE3 cells transformed with the empty plasmid pET28a. All points are expressed as the average value of 12-replicates in 4 different experiments (*n* = 4) ± SDEV. All points are statistically significant and significantly different from each other (*p* < 0.01). Asterisks are omitted for display purposes. Accession numbers for each protein used in this study can be found in Appendix A.

**Figure 6 ijms-24-09784-f006:**
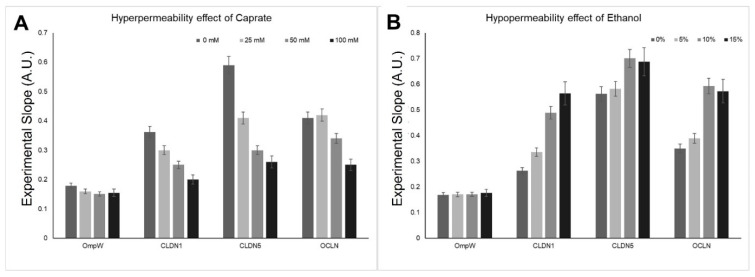
Experimental slopes of 4-α-helical CAMs in the presence of paracellular permeability modulators. Protein expression of 4-α-helical CAMs (CLDN1, CLDN5, and OCLN), or OmpW alone, is presented here. The Flow Cytometry analysis is plotted here to demonstrate the dose-dependent effect of Caprate (**Panel A**), a known agent to disrupt paracellular permeability. (**Panel B**) displays the dose-dependent effect of Ethanol, a known agent to increase tightness of the paracellular space. Data presented here was statistically analyzed. All points are expressed as the average value of 12-replicates in 4 different experiments (*n* = 4) ± SDEV. All points are statistically significant and significantly different from each other at all concentrations of the modulator (*p* < 0.01). Asterisks are omitted for display purposes.

**Figure 7 ijms-24-09784-f007:**
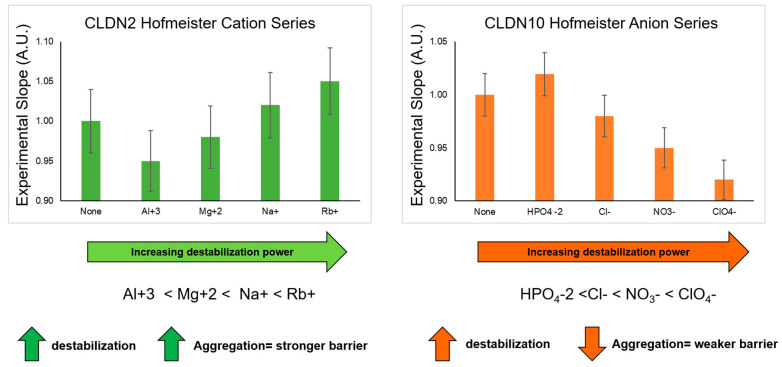
The Hofmeister series and ion permeable CLDNs. Ion permeable human CLDN2 (cations) and human CLDN10 (anions) were exposed to the influence of different salts. For CLDN2, Chloride salts were used. For CLDN10, Sodium salts were used. The left panel indicates the Hofmeister disruptive effects of cations. Similarly, the right panel displays the Hofmeister ranking of disruptive effects of the negative ions. The Flow Cytometry analysis enables the determination of Experimental Slopes that represent adhesive properties of CAMs in the presence of 100 mM salts. Data presented here were statistically analyzed. All points are expressed as the average value of 12-replicates in 4 different experiments (*n* = 4) ± SDEV. All points are statistically significant and significantly different from each other (*p* < 0.01). Asterisks are omitted for display purposes.

**Figure 8 ijms-24-09784-f008:**
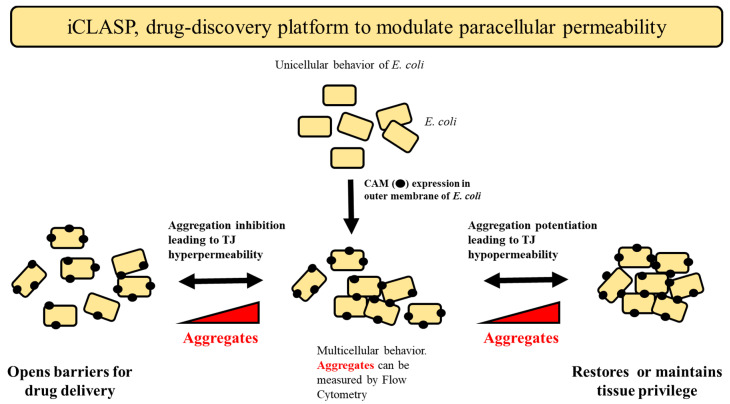
Conceptual Model. The iCLASP experimental model is presented here. Unicellular BL21 DE3 cells are transformed with plasmids hosting a fusion of OmpW and CAMs. Protein production is induced, and cells are further studied in a 96-well plate format by Flow Cytometry.

**Figure 9 ijms-24-09784-f009:**
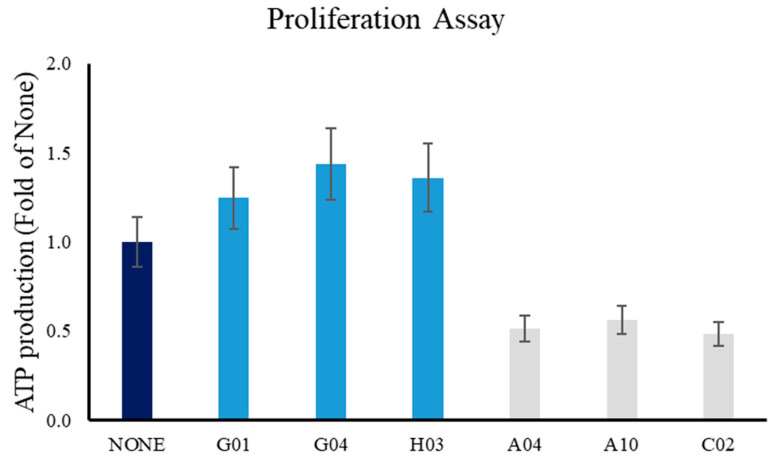
Proliferation Assay of A549 cells. Compounds identified using iCLASP were tested in a proliferation assay using A549 lung cancer cells. No compound added to A549 cells is labeled as NONE, and proliferation under these conditions was used as 100%. All other values are presented as a fold of NONE. Compounds G01, G04, and H03 increased proliferation, while A04, A10, and C02 decreased it. Commercial identifiers can be found in Appendix A. Data was collected and analyzed from four different experiments and presented as values ± STDEV. All values are different than the control (NONE) with *p* < 0.05. For visual aid, Dark Blue bar is negative control; Light Blue bars correspond to compounds that increase proliferation; Gray bars correspond to compounds that decrease proliferation.

## Data Availability

Data can be requested through the Corresponding Author.

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
