# Peer review of "Expression of Cell-Adhesion Molecules in *E. coli*: A High Throughput Screening to Identify Paracellular Modulators"

_ijms, 2023, doi:10.3390/ijms24129784_

Round 1
Reviewer 1 Report (Previous Reviewer 2)
This revised manuscript has responded appropriately to all of my suggestions for the original. The paper describes a novel molecular system to assess function epithelial Claudins and Occludin proteins of Adherens Junction and Tight Junctions. 4-helix-bundle AJ and TJ proteins are expressed in E. coli as OmpW fusions. The expressed proteins are surface expressed and mediate cell-cell adhesion of the bacteria. A simple flow cytometry light scattering assay allows for rapid assessment of adhesion properties. The procedure, named iCLASP, facilitated assessments of effects of two known perturbants of claudin function, and also some protein stability properties. In both cases, the assay results mimicked those in situ in epithelia. The bacterial system expressing CLD2 was used as primary and secondary screens for a 50,000-compound library. Six potent perturbants of bacterial adhesion were tested against A549 lung cancer cells: Three activators of adhesion increased cellular ATP, and three inhibitors decreased it, similar to the effects of cld2 knockouts in A549. In conclusion, the paper describes discovery and validation of a potentially highly useful assay for function of transmembrane 4-helix bundle proteins.
Author Response
Dear Reviewer
we appreciate your positive evaluation of our work.
Our team is excited to continue forward from this point on.
Sincerely,
Dario Mizrachi, Ph.D.
Reviewer 2 Report (New Reviewer)
The authors use iCLASP (inspection of cell-adhesion molecules aggregation through FC protocols) to identify paracellular modulators of CLDN2 (Claudin 2, one CAMs’ components of the tight junction). The authors expressed the fused CAMs (cell-adhesion molecules) in E. coli,, leading to the cells’ aggregation. Then they used the method iCLASP (inspection of cell-adhesion molecules aggregation through FC protocol) to do high throughput screening to identify small molecules which interact with CAMs and affect CAMs aggregation properties.
1. Page 10, line 2, Rb+2 should be Rb+2. Fig6. In Panel A, without Caprate, OmpW is 0.18, CLDN1 is 0.37; In Panel B, without Ethanol, OmpW is 0.17, CLDN1 is 0.26. Why is there such a big difference? Are those two experiments performed under the same condition?
3. How were the 50,000 compounds analyzed?
4. For the 6 compounds identified, can the authors show those compounds’ effect on the CLDN2’s aggregation properties?
5. Mouse experiments of those 6 compounds are suggested, as they can demonstrate the impact of their experiment methods in cancer research.
Quality of English is good.
Author Response
Dear Reviewer,
on behalf of our team we appreciate your positive review of our work.
We will address the corrections suggested.
Additionally, we will create an additional Supplementary figure to address the work flow using iCLASP that lead to the discovery of the molecules. We will include Graphs of the final compounds's effects on E coli. This new supplementary will address points 3 and 4.
Overexpression of OmpW does not affect the aggregation of E coli. That Experimental Slope is a background slope of non-aggregated E coli.
In a given graph, all experiments are run in 12-replicates/plate and 4 plates per day, corresponding to 48 replicates of each condition. Each plate contained cells alone, cells overexpressing OmpW and cells overexpressing OmpW-Claudin plasmids.
For the 50,000 compounds, the control was OmpW-Claudin 2 without treatments.
Animal studies are part of the future of this research. We hope that in the next year, we will have results for a manuscript.
Thank you once again.
Sincerely,
Dario Mizrachi, Ph.D.
This manuscript is a resubmission of an earlier submission. The following is a list of the peer review reports and author responses from that submission.
Round 1
Reviewer 1 Report
This manuscript reports a series of experiments to generate and validate the use of a bacterial model for evaluation of drugs that alter tight junction-dependent paracellular permeability. The major flaw of this work is the underlining assumption that claudin function as paracellular barrier/pore can be equated with or is a measure of its (very weak) adhesive function. Considering the fact that the Claudine sequences involved in cis and trans-interactions are distinct from those involved in paracellular barrier/pore formation, this assumption is not justified. Thus, a screen for molecules that affect claudin-dependent adhesion has little relevance as a screen for molecules that affect barrier function, especially in a completely artificial context, that lacks the complex organization (anchoring, interaction with specific transmembrane proteins and lipids, etc) of vertebrate epithelial and endothelial cells. A second major flaw is the underlying assumption that TJ transmembrane proteins expressed as fusions with a bacterial protein represent a reliable model, with structural and functional characteristics similar to those occurring in vertebrate epithelial and endothelial tissues. No evidence is presented for example that claudin cis and trans-interactions occur, that claudins are polymerized into intramembrane fibril structures, etc. Such polymerization requires scaffolding by ZO proteins (Umeda et al, Cell 2006), which are not expressed in bacteria. It is also unclear why proliferation (viability) of cancer cells is used to test the activity of compounds isolated by the iCLASP technique, when no evidence is presented about the role of claudin-2 in this process, or about a mechanistic link between claudin-2 function as an adhesion molecule and paracellular pore and as a regulator of cell proliferation.
Specific points.
Figure 1C. The control (-IPTG) appears to have many fewer cells than the + IPTG sample. Possibly a series of images at increasing dilutions of the same number/density of initial bacterial cultures could be more informative.
Figure 3. It is worrying that the “control” protein, the AMPA receptor adaptor protein STRGZ, shows a similar experimental slope as most of the claudins. This indicates that none of the changes in experimental slopes are significant, and confirms that claudins, occludin and tricellulin are not important determinants of cell-cell adhesion.
Figure 4. The use of non-specific agents (detergents, ethanol) that are known to affect TJ permeability, to support the idea that any agent that disrupts permeability will also affect adhesion is not justified.
Figure 5. How ions that are supposed to go through the paracellular channel on the basis of their size/charge would affect cell-cell adhesion is not clear. Cell-cell adhesion is not equivalent to or a measure of barrier function.
Figure 7. The ATP assay is a cell viability assay, not a cell proliferation assay.
Figure 7. No evidence (siRNA-dependent rescue of phenotype, etc) is shown that altered viability of cells treated with different compounds actually depends on claudin-2, or whether the compounds do affect claudin-2 expression, localization, etc.. The compounds could act through many other pathways.
Author Response
Dear Reviewer
As the corresponding author I thank you for your thoughtful discussion. The iCLASP technique does not seek to quantify the strength of individual proteins under the present experimental design. The inclusion of controls in each experiment, Cells only, Cells expressing OmpW or cpOmpW alone, serve as parameters to observe that cell adhesion molecules exert an effect in the unicellular behavior of E. coli. The observation that cells aggregate came as a result of reading the literature. Initially our efforts focused on perhaps replicating trans-epithelial electrical resistance using E. coli. Bonander et al. [1] overexpressed claudin-1 in yeast and observed how these cells also aggregated. They qualified the aggregates using Flow Cytometry. We further the design by creating fusions of OmpW and CAMs to express in the outer membrane of E. coli, and further quantify the aggregates by calculating the slopes.
Initially we worked with claudins since they are not glycosylated and the platform suited that property. We further discuss how iCLASP might be limited to study certain glycosylated CAMs in the discussion.
Our technique distinguishes cellular behavior, unicellular or multicellular. Claudins are recognized as the most relevant membrane proteins in the TJ responsible for permeability[2]. In a recent publication by our group, we demonstrate that claudin-1 homotypic interactions are close to 1,000 fold stronger than epithelial cadherin (E-CAD) self-association even in the presence of calcium [3,4]. Ours is the first biophysical characterization of these interactions. Again, our strategy was to use E-CAD as a control and not to determine absolute values. Nevertheless, our determinations of affinity for E-CAD were within 2-fold of literature reported values. We have encountered in the literature the two camps as to which junctions are stronger, Tight Junctions or Adherens Junctions. One encyclopedic article in Nature hints that the TJs are very strong: “Tight junctions form a seal between cells that is so strong that not even ions can pass across it.” (https://www.nature.com/scitable/topicpage/cell-adhesion-and-cell-communication-14050486/). Again, there has been no point of comparison with recombinant proteins up until our reports last year.
Perhaps due to our simplistic approach in this first report using iCLASP we indeed may not have addressed all of the properties of CAMs, specifically claudins. As the reviewer indicated, a great role for intracellular proteins in the TJ and AJ polarity and assembly is highly relevant[5,6]. Our technique only concerns with cell-adhesion, this has been shown to reside in the extracellular domains of claudins[7,8] and JAM-A[9]. A goal for future work is produce intracellular scaffolding in E. coli alongside iCLASP to attempt polarization of cells.
The central role of iCLASP is to provide a reliable way to measure effects of small molecules in adhesion. Cell-adhesion proteins like E-CAD are calcium dependent. Cis-interactions of E-CAD are favored in low calcium environments like the intracellular milieu. It is believed that claudins also traffic to the plasma membrane in an “inactive” form, cis-interactions. Nevertheless, claudins are not calcium dependent, and upon reaching the plasma membrane, they engage in cell-cell interactions (trans). E-CAD depends on extracellular high calcium concentrations to achieve trans- or cell-cell interactions. We provide literature to explain why STARGAZIN (AMPA co-receptor) is expected to have adhesive properties. As stated above, iCLASP under the present conditions cannot distinguish precisely the differences between the strength of adhesion. It could be addressed at a later time by normalizing adhesion and protein expression.
The iCLASP idea was born due to the lack of tools to identify paracellular permeability modifiers. Current techniques, TEER, are low-throughput and unreliable in many cases[10]. Other bottlenecks for the study of cell-adhesion includes the complexity of junctions, with multiple intracellular components and membrane proteins. The lack of cellular models to independently study individual claudins for example, of the tissue-culture available cell lines not all have tight junctions of a given composition. Finally, if a cell line is identified for the adequate study of paracellular permeability the TEER values achieved in a homogenous monolayer may be inadequate (too low). A study presented data that claudin-2 overexpression enhances tight junction permeability to small cations but overall paracellular permeability decreased[11]. Claudin-2 overexpression in cancer is considered pro-metastasis. Considering the number of reports linking claudin-2 to proliferation in A549 cells proceeded to use that as a measurement of confidence for the small molecules identified by iCLASP bacterial system. In our laboratory TEER values in A549 were extremely low and a difference in permeability as a result of TEER values could not be achieved with confidence. This observation is also present the literature for A549 cells[12].
We remain appreciative of your efforts and constructive review.
Sincerely,
Dario Mizrachi, Ph.D.
Tuesday, January 11, 2022
Specific points.
Figure 1C. The control (-IPTG) appears to have many fewer cells than the + IPTG sample. Possibly a series of images at increasing dilutions of the same number/density of initial bacterial cultures could be more informative. Unfortunately, E. coli was difficult to adhere to the grids, even at different more higher concentrations. We feel confident that the intent of qualitatively demonstrate the difference between unicellular and multicellular behavior is achieved. No further conclusions are derived from Figure 1.C.
Figure 3. It is worrying that the “control” protein, the AMPA receptor adaptor protein STRGZ, shows a similar experimental slope as most of the claudins. This indicates that none of the changes in experimental slopes are significant, and confirms that claudins, occludin and tricellulin are not important determinants of cell-cell adhesion. We have addressed with a reference to the possibility STRGZ is more than a homolog to claudins but also may contain adhesion information. A more recent article addresses the possibility that STRGZ may act as a partner to aid AMPA achieve oligomeric states needed for acvtivity[13].
Figure 4. The use of non-specific agents (detergents, ethanol) that are known to affect TJ permeability, to support the idea that any agent that disrupts permeability will also affect adhesion is not justified. We have addressed with references for each case in the manuscript.
Figure 5. How ions that are supposed to go through the paracellular channel on the basis of their size/charge would affect cell-cell adhesion is not clear. Cell-cell adhesion is not equivalent to or a measure of barrier function. This is a valid question for which we offer no answer, yet we show that claudin 2, a cation permeable claudin, is affected by cations; and the opposite to claudin 10, affected by anions. Adhesion may be equated to flexibility to make the TJ permeable. There is a lack of evidence in the literature as to the effect of ions in the paracellular permeability. We cite again the claudin-2 case where its overexpression leads to cation permeability but overall permeability decrease.
Figure 7. The ATP assay is a cell viability assay, not a cell proliferation assay.
Proliferation is an increase in the number of cells as a result of cell growth and cell division. ATP bioluminescence as a measure of cell proliferation and cytotoxicity has been reported[14]. Perkinelmer (https://www.perkinelmer.com/product/atp-lite-300-assay-kit-6016943) product is suitable for proliferation and cytotoxicity assays.
Figure 7. No evidence (siRNA-dependent rescue of phenotype, etc) is shown that altered viability of cells treated with different compounds actually depends on claudin-2, or whether the compounds do affect claudin-2 expression, localization, etc. The compounds could act through many other pathways. We present no further evidence rather than the citations we included in the manuscript.
References
- Bonander, N.; Jamshad, M.; Oberthur, D.; Clare, M.; Barwell, J.; Hu, K.; Farquhar, M.J.; Stamataki, Z.; Harris, H.J.; Dierks, K.; et al. Production, purification and characterization of recombinant, full-length human claudin-1. PLoS One 2013, 8, e64517, doi:10.1371/journal.pone.0064517.
- Gunzel, D.; Yu, A.S. Claudins and the modulation of tight junction permeability. Physiol Rev 2013, 93, 525-569, doi:10.1152/physrev.00019.2012.
- Mendoza, C.; Nagidi, S.H.; Collett, K.; McKell, J.; Mizrachi, D. Calcium regulates the interplay between the tight junction and epithelial adherens junction at the plasma membrane. FEBS Lett 2021, doi:10.1002/1873-3468.14252.
- Taylor, A.; Warner, M.; Mendoza, C.; Memmott, C.; LeCheminant, T.; Bailey, S.; Christensen, C.; Keller, J.; Suli, A.; Mizrachi, D. Chimeric Claudins: A New Tool to Study Tight Junction Structure and Function. Int J Mol Sci 2021, 22, doi:10.3390/ijms22094947.
- Drees, F.; Pokutta, S.; Yamada, S.; Nelson, W.J.; Weis, W.I. Alpha-catenin is a molecular switch that binds E-cadherin-beta-catenin and regulates actin-filament assembly. Cell 2005, 123, 903-915, doi:10.1016/j.cell.2005.09.021.
- Rouaud, F.; Sluysmans, S.; Flinois, A.; Shah, J.; Vasileva, E.; Citi, S. Scaffolding proteins of vertebrate apical junctions: structure, functions and biophysics. Biochim Biophys Acta Biomembr 2020, 1862, 183399, doi:10.1016/j.bbamem.2020.183399.
- Hagen, S.J. Non-canonical functions of claudin proteins: Beyond the regulation of cell-cell adhesions. Tissue Barriers 2017, 5, e1327839, doi:10.1080/21688370.2017.1327839.
- Lim, T.S.; Vedula, S.R.; Hunziker, W.; Lim, C.T. Kinetics of adhesion mediated by extracellular loops of claudin-2 as revealed by single-molecule force spectroscopy. J Mol Biol 2008, 381, 681-691, doi:10.1016/j.jmb.2008.06.009.
- Prota, A.E.; Campbell, J.A.; Schelling, P.; Forrest, J.C.; Watson, M.J.; Peters, T.R.; Aurrand-Lions, M.; Imhof, B.A.; Dermody, T.S.; Stehle, T. Crystal structure of human junctional adhesion molecule 1: implications for reovirus binding. Proc Natl Acad Sci U S A 2003, 100, 5366-5371, doi:10.1073/pnas.0937718100.
- Mukherjee, T.; Squillantea, E.; Gillespieb, M.; Shao, J. Transepithelial electrical resistance is not a reliable measurement of the Caco-2 monolayer integrity in Transwell. Drug Deliv 2004, 11, 11-18, doi:10.1080/10717540490280345.
- Weber, C.R.; Liang, G.H.; Wang, Y.; Das, S.; Shen, L.; Yu, A.S.; Nelson, D.J.; Turner, J.R. Claudin-2-dependent paracellular channels are dynamically gated. Elife 2015, 4, e09906, doi:10.7554/eLife.09906.
- Radiom, M.; Sarkis, M.; Brookes, O.; Oikonomou, E.K.; Baeza-Squiban, A.; Berret, J.F. Pulmonary surfactant inhibition of nanoparticle uptake by alveolar epithelial cells. Sci Rep 2020, 10, 19436, doi:10.1038/s41598-020-76332-7.
- Shaikh, S.A.; Dolino, D.M.; Lee, G.; Chatterjee, S.; MacLean, D.M.; Flatebo, C.; Landes, C.F.; Jayaraman, V. Stargazin Modulation of AMPA Receptors. Cell Rep 2016, 17, 328-335, doi:10.1016/j.celrep.2016.09.014.
- Crouch, S.P.; Kozlowski, R.; Slater, K.J.; Fletcher, J. The use of ATP bioluminescence as a measure of cell proliferation and cytotoxicity. J Immunol Methods 1993, 160, 81-88, doi:10.1016/0022-1759(93)90011-u.
Reviewer 2 Report
This manuscript describes a surface display model in E. coli and its use to investigate homophilic interactions among mammalian cell adhesion molecules involved in Tight Junctions (TJ-CAMS) including claudins (CLDN), occludins (OCLDN) and hJAM-A. The method relies on fusions to outer membrane proteins or their circularly-permuted forms. Surface display of any of 13 tested TJ-CAMS leads to increased bacterial aggregation. The manuscript also describes similarity in effects of known TJ perturbing agents on the aggregation of the bacteria. The utility of the model is demonstrated in a high-throughput screen that identifies 6 novel perturbants of CLDN2-mediated bacterial aggregation. These compounds affect proliferation of lung carcinoma cells as predicted based on data from overexpression or repression of CLDN2. That is: overexpression or addition of the CLDN2 activators increased proliferation and CLDN2 inhibitors and repressors decreased cell proliferation.
On the whole, the manuscript is clear and compelling, and therefore this is a desctiption of a new tool likely to be highly useful tool for investigations of TJs and the blood-brain barrier. Two points need to be clarified, however.
- The E. coli surface display system cannot glycosylate CAMs, so all interactions are of non-glycosylated forms. This limitation should be acknowledged.
- Error bars are reported as s.e.m. for 4 independent biological replicates of 12 trials each. If s.e.m is calculated for N=48, then its value is suspect. If N=4, then it is indeed a useful measure of variance. The method of calculation should be explicit, and examples of s.d. for technical replicates and for the independent determinations should also be given.
Edits and minor corrections:
- 2: The following language should be restructured and clarified. The second and third sentences seem contradictory, and the last sentence lacks elements of what is being combined.
Controlling the TJ is a strategy for the manipulation of the paracellular permeability[
26]. The development and delivery of small molecule drugs is relatively straightforward. Drug discovery of small molecules from target selection through to clinical evaluation is a very complex and challenging area of study. The main obstacle in drug discovery is the initial hit-finding[30,31]. For example, the combination of so little effort in developing solutions to the BBB permeability leads directly to the present situation in neurotherapeutics, few effective treatments for most brain-related disorders.
- Legend for Fig. 2 should be “…the latter is larger…” [double t in latter].
- Legend for Fig. 2: Larger aggregates do not necessarily mean stronger protein-protein interactions. Increased aggregate size also can reflect differences in expression levels or aggregation kinetics.
- p. 6: “…the effects of slats that have been well characterized…” should be “salts”
- p. 8: Section 2.5 “From the second scree,…” should be “screen”
- p. 8: Section 2.5: The change in slope for hyper-aggregators seems underwhelming at 8-15%. It would be helpful to add some of the appropriate screen data to supplemental data.
- Fig. 8: The model will be clearer if the labels “Hyperpermeability” and “Hypopermeability” were modified to say “Aggregation inhibition leading to TJ hyperpermeability” and “Aggregation potentiation leading to TJ hypopermeability.”
- p. 12 section 4.4: ATCC number is missing.
Author Response
Dear Reviewer
As the corresponding author, I thank you for your thoughtful discussion. The iCLASP technique does not seek to quantify the strength of individual proteins under the present experimental design. The inclusion of controls in each experiment, Cells only, Cells expressing OmpW or cpOmpW alone, serve as parameters to observe that cell adhesion molecules exert an effect in the unicellular behavior of E. coli. The observation that cells aggregate came as a result of reading the literature. Initially our efforts focused on perhaps replicating trans-epithelial electrical resistance using E. coli. Bonander et al. [1] overexpressed claudin-1 in yeast and observed how these cells also aggregated. They qualified the aggregates using Flow Cytometry. We further the design by creating fusions of OmpW and CAMs to express in the outer membrane of E. coli, and further quantify the aggregates by calculating the slopes.
Below we address, red font, your concerns and suggestions.
We remain appreciative of your constructive review.
Sincerely,
Dario Mizrachi, Ph.D.
Tuesday, January 11, 2022
On the whole, the manuscript is clear and compelling, and therefore this is a desctiption of a new tool likely to be highly useful tool for investigations of TJs and the blood-brain barrier. Two points need to be clarified, however.
- The E. coli surface display system cannot glycosylate CAMs, so all interactions are of non-glycosylated forms. This limitation should be acknowledged. Was addressed in the discussion.
- Error bars are reported as s.e.m. for 4 independent biological replicates of 12 trials each. If s.e.m is calculated for N=48, then its value is suspect. If N=4, then it is indeed a useful measure of variance. The method of calculation should be explicit, and examples of s.d. for technical replicates and for the independent determinations should also be given. Was addressed.
Edits and minor corrections:
- 2: The following language should be restructured and clarified. The second and third sentences seem contradictory, and the last sentence lacks elements of what is being combined.
Controlling the TJ is a strategy for the manipulation of the paracellular permeability[
26]. The development and delivery of small molecule drugs is relatively straightforward. Drug discovery of small molecules from target selection through to clinical evaluation is a very complex and challenging area of study. The main obstacle in drug discovery is the initial hit-finding[30,31]. For example, the combination of so little effort in developing solutions to the BBB permeability leads directly to the present situation in neurotherapeutics, few effective treatments for most brain-related disorders.
** We included new references and improved the language to aid the reader in understanding the principle that most neuro pharmaceuticals are administered taking advantage of the transcellular route and only small molecules (<400 Da) cross the paracellular space. Intentional drug discovery would occur if the paracellular barrier could be controlled.
- Legend for Fig. 2 should be “…the latter is larger…” [double t in latter]. Corrected
- Legend for Fig. 2: Larger aggregates do not necessarily mean stronger protein-protein interactions. Increased aggregate size also can reflect differences in expression levels or aggregation kinetics. All experiments are compared to internal controls, cells only, cells expressing OmpW, and OmpW-CAM.
- p. 6: “…the effects of slats that have been well characterized…” should be “salts” Corrected
- p. 8: Section 2.5 “From the second scree,…” should be “screen” Corrected
- p. 8: Section 2.5: The change in slope for hyper-aggregators seems underwhelming at 8-15%. It would be helpful to add some of the appropriate screen data to supplemental data. The changes in paracellular permeability measured by TEER to describe hyperpermeability when compared to hypermeability are very different. Decreasing claudin-claudin interactions result in higher effects than the opposite for the same cell type[2]. Increasing cell-cell interactions relies on existing CAM concentrations of proteins that might already be engaged in cell-adhesion. There is no opportunity for cells to express CAMs further to produce greater extent of aggregation.
- Fig. 8: The model will be clearer if the labels “Hyperpermeability” and “Hypopermeability” were modified to say “Aggregation inhibition leading to TJ hyperpermeability” and “Aggregation potentiation leading to TJ hypopermeability.” We have adopted this suggestion nand made the changes to Figure 8
- p. 12 section 4.4: ATCC number is missing. Corrected the omission.
References
- Bonander, N.; Jamshad, M.; Oberthur, D.; Clare, M.; Barwell, J.; Hu, K.; Farquhar, M.J.; Stamataki, Z.; Harris, H.J.; Dierks, K.; et al. Production, purification and characterization of recombinant, full-length human claudin-1. PLoS One 2013, 8, e64517, doi:10.1371/journal.pone.0064517.
- Kuzmanov, I.; Herrmann, A.M.; Galla, H.J.; Meuth, S.G.; Wiendl, H.; Klotz, L. An In Vitro Model of the Blood-brain Barrier Using Impedance Spectroscopy: A Focus on T Cell-endothelial Cell Interaction. J Vis Exp 2016, doi:10.3791/54592.

Round 2
Reviewer 1 Report
Major flaws in design and interpretation of experiments. No satisfactory response to comments.